# Epidemiology of Injuries in First Division Spanish Women’s Soccer Players

**DOI:** 10.3390/ijerph18063009

**Published:** 2021-03-15

**Authors:** Rodrigo Martín-San Agustín, Francesc Medina-Mirapeix, Andrea Esteban-Catalán, Adrian Escriche-Escuder, Mariana Sánchez-Barbadora, Josep C. Benítez-Martínez

**Affiliations:** 1Department of Physiotherapy, University of Valencia, 46010 Valencia, Spain; Rodrigo.martin@uv.es (R.M.-S.A.); aeses3@alumni.uv.es (A.E.-C.); sanbarm2@alumni.uv.es (M.S.-B.); josep.benitez@uv.es (J.C.B.-M.); 2Department of Physiotherapy, University of Murcia, 30100 Murcia, Spain; mirapeix@um.es; 3Department of Physiotherapy, University of Malaga, 29010 Malaga, Spain

**Keywords:** epidemiology, female athlete, first division, soccer, injury incidence

## Abstract

The epidemiology of injuries in female soccer has been studied extensively in several national leagues. Even so, data on the first division Spanish league are limited. The objective of this study was to describe the epidemiology of the first division of the Spanish Women’s Soccer League and to analyze data in relation to game position, circumstance, or the moment of injury. Fifteen teams and 123 players participated in the study. Players’ characteristics and their injuries (location, type, diagnosis, circumstance, and moment) were collected. Injuries were described by their frequencies (number and percentage) and incidence rates (IR) with 95% confidence intervals (CIs). Lower limb injuries accounted for 86.8% of total injuries. Anterior cruciate ligament (ACL) and meniscus injuries occurred in totality in non-contact circumstance (0.35/1000 h; 95% CI, 0.18 to 0.62 and 0.23/1000 h; 95% CI, 0.10 to 0.45, respectively). Match injury IRs (19.02/1000 h; 95% CI, 14.89 to 23.97) were significantly higher than training (1.70/1000 h; 95% CI, 1.27 to 2.22). As a conclusion, structures such as the ACL or meniscus are most commonly injured in the non-contact circumstance in the first division of the Spanish Women’s Soccer League. In addition, match situations involve a greater risk of injury than training, increasing the risk to the ankle and knee injuries as the season progresses.

## 1. Introduction

Around 29 million soccer players in the world are female, representing 10% of the total. In addition, the numbers are constantly growing with an estimated 51% increase in the number of registered female players (at young and senior level) from 2000 to 2006 (FIFA communications division). There is also an increase in the interest of society, and specifically in Spain, since the first division Spanish women’s soccer players has established record values of spectators within its league in the 2016/2017 season with 260,000 televiewers for a match and 22,202 attendees to a soccer stadium [1]. This league is formed by sixteen professional teams; it takes place from September to May, and it has the fourth best coefficient according to UEFA, just behind the French, German, and English leagues. Thus, it is different from other major leagues in terms of the number of teams, calendar, and winner’s award system [2,3,4,5].

Although soccer has reported cardiovascular and bone health benefits [6], injuries are frequent in females [3,4,5] and can increase the risk of knee osteoarthritis in previously injured females [7]. Previous studies have shown high incidences of injury for elite female soccer players both at a young age [8,9] and in the first leagues seniors [3,4,5,10]. Thus, injury incidence rates (IRs) have been described in a range between 1.1 to 3.1 injuries per 1000 training hours and 12.6 to 23.6 injuries per 1000 match hours taking into account the Swedish [2], German [3], Norwegian [5], or American senior first leagues [4].

The first step in prevention is to identify and describe the IRs [11] and to identify priorities for injury prevention. Even so, the only existing data on injury incidence related to the first division Spanish women’s soccer players is in relation to a single club [12]. Otherwise, specific injury incidence at different periods of the season and whether these injuries occurred in training or matches, which are considered determinants of sport behavior, Ref. [13] have not been examined by previous research in other leagues [2,3,4,5]. Thus, the exact analysis of incidence rates considering these determinants, together with other factors that can influence the rates such as game position or circumstance, is essential for developing prevention programs as well as for evaluating prophylactic measures [14]. Therefore, the aim of this study was to examine the injury frequencies and IRs of the first division Spanish women’s soccer players and to analyze data in relation to different variables—game position, circumstance (i.e., contact/non-contact), training/match, and the period of the season to direct the purpose of injury prevention strategies.

## 2. Materials and Methods

This study used a descriptive epidemiology design for the injuries registered retrospectively that occurred during the 2016/2017 season in the first division Spanish women’s soccer players.

We contacted and invited all 16 teams of the first division Spanish women’s soccer players that competed in the 2016/2017 season (30 league matches) to participate in this retroactive study at the end of the season. To contact each team, the contact information of each team captain was obtained by a researcher in this study related to Spanish women’s football. After explaining the study via telephone, captains requested permission from their respective clubs to be part of it. Only one club prevented the participation of its players in the study for reasons related to confidentiality. The players who had consent from their clubs were contacted to fill out the informed consent. A total of 15 teams and 123 players (a mean of eight players per team; a range from 5 to 14) participated in this study, representing 44% of the total players in the league and with a mean age of 23.27 (4.07) years.

### 2.1. Procedures

A retrospective survey was developed to examine players’ characteristics (age, anthropometric data, and game position), their injuries (location, type, diagnosis, the circumstance of the injuries, and match/training), and the period of the season the injuries occurred, using a questionnaire. The questionnaire used was the one proposed by Fuller et al. [15] adapted to the Google Forms format. At the beginning of the questionnaire, it was emphasized that only those injuries diagnosed by a physical therapist or doctor were included and the medical staff would be consulted in case of doubt for any of the items asked.

Otherwise, for the exposure times, the coaching staff of three individual teams was contacted prospectively at the beginning of the season. It was requested that they collect the exposure times by each season period for training/match of each player. This study and its respective questionnaire were approved by the Ethics Committee of the University of Valencia, Valencia, Spain (H1507224419757).

### 2.2. Exposure Times (by Season’s Period and Training/Match)

We analyzed three entire teams individually to obtain the means of exposure hours to training and the match for each season period [16] and to estimate the exposure times of the other teams. The exposure hours were calculated by dividing the minutes by 60.

The season was divided into the preseason (July and August), start of the season (September, October, and November), mid-season, (December, January, and February) and end of the season (March, April, May, and June). The number of matches corresponding to the league per period was 11 for the start of the season, nine for the mid-season, and ten for the end of the season.

On the one hand, training exposure was calculated by multiplying the duration (in minutes) of every training session by the players who had participated in each session (per team). Injured players were not counted. Moreover, the warm-up period before matches (25 min) was also considered as training exposure, multiplied by the number of players that participated in each match (11 starters and the substitutes who played later). Otherwise, the match exposure was calculated considering the individual minutes that each player played both in the league, other tournaments, or friendly matches [12]. The hours of exposure were calculated by dividing the minutes by 60.

### 2.3. Location, Type, Diagnosis, Circumstance, Training/Match and Season’s Period

According to the consensus statement on injury definitions [15], an injury is defined as any physical complaint sustained by a player that results from a soccer match or training, irrespective of the need for medical attention. We used this statement for the location and type of the injuries, which has been used by previous authors [12,16]. In this way, and as the statement suggests, we grouped locations and types with small frequencies, using eight locations in total (hip/groin, thigh, knee, lower leg, ankle, foot/toe, upper limb and, trunk/pelvis) and three types of injury (joint/ligament, muscle/tendon, and fracture/bone stress). In addition, the specific diagnosis of each injury was also collected (e.g., ligament injuries subdivided into distension, partial rupture, and total rupture) and the circumstance of each injury was detailed according to whether it had occurred due to contact with a player or object (contact) or not (non-contact) [12,15]. Finally, the injuries were also classified according to whether they occurred in match or training and depending on the period of the season.

### 2.4. Statistical Analysis

The characteristics of the participants and exposure times (by the period of the season and training/match) were expressed as means and standard deviations (SD). Injuries were expressed as frequencies (number and %) and IRs (per 1000 h of exposure) with their 95% confidence intervals (CIs). Significant differences between IRs were calculated if there was no overlap between their 95% CIs, with a level of significance *p* < 0.05 [17]. To reduce the probability of recording false positives, in the cases of multiple comparisons (e.g., type of lesion), we calculated the 99% CI to evaluate their overlap, increasing the acceptance level from 0.05 to 0.01. In addition, risk ratios (RRs) were also calculated to assess the effect size, as $\mathrm{RR}={\mathrm{IR}}_{1}/{\mathrm{IR}}_{2}$ [18].

## 3. Results

### 3.1. Player Characteristics and Exposure Time

Players’ characteristics and exposure time, as a whole and stratified by game position (13 goalkeepers, 40 defenders, 45 midfielders, and 25 forwards), are shown in Table 1. The mean age of the players was 23.27 ± 4.07 years, with a first division debut age of 18.01 ± 2.77 years, a body mass of 59.5 ± 5.7 kg, a stature of 164.9 ± 5.7 cm and, a BMI of 21 to 62 ± 1.56 kg/m^2^. On the other hand, the total (251.57 ± 43.75 h), training (222.50 ± 36.73 h), and match (29.07 ± 13.69 h) exposure times are also shown by the position and period of the season. These exposure times were used for the subsequent calculation of IRs. No differences were observed between game positions for any of the characteristics or exposure times.

The frequencies and IRs of injuries according to training/match, type, and circumstances are shown in Table 2 in the aggregate and by position. The total injury IR was 3.65/1000 h (95%CI = 3.02, 4.37). The whole of the sample had a significantly higher IR in matches (19.02/1000 h; 95%CI = 14.89, 23.97) versus training (1.70/1000 h; 95%CI = 1.27, 2.22), with an RR of 11.2 and significant differences also by game positions. Regarding the type of injury, joint/ligament injuries were the type that had the highest IRs (2.07/1000 h; 95%CI = 1.61, 2.62) followed by muscle/tendon (1.23/1000 h; 95%CI = 0.88, 1.67) for all of the players but without significant differences between them. In contrast, the injury IRs of fracture/bony stress were significantly less than the other injury types. Concerning the game position, fracture/bone stress injuries had significantly lower IRs than joint/ligament injuries for all positions. Finally, non-contact injuries were more frequent than contact injuries for all players (2.20/1000 h; 95%CI = 1.72, 2.77 vs. 1.45/1000 h; 95%CI = 1.07, 1.93; RR of 1.5) and across all positions, although the difference was not significant.

### 3.2. Injury Location

The frequencies and IRs of locations and diagnosis of injuries by circumstance are shown in Table 3. Lower limb injuries accounted for 86.8% of the total injuries. Regardless of the circumstance; ankle (0.97/1000 h; 95%CI = 0.66, 1.37), knee (0.94/1000 h; 95%CI = 0.64, 1.32), and thigh (0.78/1000 h; 95%CI = 0.51, 1.13) showed greater IRs than the rest of the locations (less than 0.32/1000 h). The most frequent diagnoses were LCL distension, ACL total rupture, and hamstring strain. Among these diagnoses, only ACL rupture had a significantly higher IR in non-contact than contact circumstance.

Within non-contact injuries, thigh and knee injuries accounted for 57.4% of the total injuries, versus injuries with contact, where these injuries only represented 31.1%. The ankle was the most injured structure with contact with 42.2%. Regarding ankle injuries, the circumstance with contact (0.61/1000 h; 95%CI = 0.38, 0.94) had an RR of 1.7 compared with non-contact (0.35/1000 h; 95%CI = 0.18, 0.62). Considering injuries of knee structures, all ACL and meniscus injuries were non-contact (0.35/1000 h; 95%CI = 0.18, 0.62 and 0.23/1000 h; 95%CI = 0.10, 0.45, respectively). In contrast, MCL injuries showed an RR of 2.5 in the circumstance with contact (0.26/1000 h; 95%CI = 0.12, 0.49) versus non-contact (0.10/1000 h; 95%CI = 0.20, 0.26). Finally, thigh injuries had an RR of 3.1 comparing non-contact (0.58/1000 h; 95%CI = 0.35, 0.90) than with contact (0.19/1000 h; 95%CI = 0.08, 0.40). However, only ACL and meniscus injuries showed significant differences between circumstances.

### 3.3. Incidences by the Season’s Period

The frequencies and IRs of injuries of the lower limb depending on the period of the season and whether they were training or in the match are shown in Table 4. Lower limb injury IRs in the match had higher significant values in all periods with respect to training, with RRs of 4 in the start season and 34.2 by the end of it.

Regarding ankle injuries, IRs in the last two periods had higher significant values in the match (range between 4.53 and 5.03/1000 h) compared to training (range between 0.11 and 0.71/1000 h). Knee injuries had similar behavior since IRs in the last two periods had higher significant values in the match (range between 7.25 and 10.05/1000 h) versus training (range between 0.11 and 0.29/1000 h). Finally, thigh injuries had a greater significant IR in the match than training in the preseason and at the end of the season (9.07/1000 h; 95%CI = 3.32, 20.11 versus 0.99/1000 h; 95%CI = 0.25, 2.70 (RR = 9.2 and 2.72/1000 h; 95%CI = 0.69, 7.40 versus 0.11/1000 h; 95%CI = 0.01, 0.55 (RR = 24.7), respectively).

## 4. Discussion

According to our knowledge, our study is the first to examine the frequencies and IRs of injuries in the first division Spanish women’s soccer players. We found three main findings: first, there were no differences between game positions in the injury IRs; second, the ankle injuries had greater IR in contact circumstance and the thigh and knee injuries had higher IRs in non-contact; and third, the injury IRs in match were 11 times greater than in training in total injuries. This difference increased at the end of the season for lower limb injuries.

To our knowledge, our study is the first to report injury IRs based on the game position in female soccer. Our findings regarding the non-existence of differences in injury IRs between game positions are consistent with previous studies carried out in male leagues [19]. Even so, our findings are partially unexpected since both the female leagues [20] and the male leagues [21] have described different game functional demands among game positions. For example, female forwards perform more distance at high speed in a match than the other positions [20], whilst the male forward is the position that performs more actions with contact in the English Premier League [21]. Since there are different functional demands between game positions in others leagues [19,20], we expected to find differences in our study between injury IRs between positions. Even so, an analysis within the Spanish Women’s Soccer League of such functional demands is necessary to be compared with our findings of IRs.

Regarding the location of the injuries, most of them occurred in the lower limb as described by other authors [5,10]. Although in the literature there are differences between what is the most injured structure [3,4,5,10], our findings are similar to those found in the Norwegian [5] and Swedish [10] leagues, with a higher number of injuries of the ankle compared to the knee and the thigh. Besides being the most injured structure, ankle injuries had a higher IR of contact than non-contact, being consistent with previous studies [19]. This finding has been explained previously as a result of inadequate rehabilitation routines leading to re-injuries [5] and also given that the ankle is a structure that receives many contact actions [19]. To solve this, prevention strategies, based on neuromuscular training (static and dynamic stretching, strength, agility, and balance components), have been proposed [22], which have been shown to reduce the incidence of ankle injuries. These strategies should be included in the Spanish league, with special emphasis on exercises that require contact with partners.

The differences observed between the match and training IRs on the ankle and the knee in the last two periods could be explained by the increase in game intensity (without increasing exposure time) due to the play-off or tournament positions [4], the cumulative fatigue [23] and the deterioration of the physical condition [24] as the season progresses. Contrary to what happens in other leagues, which have a break in the middle of the competition period, during which they perform a second preparatory period, this does not happen in the first division Spanish women’s soccer players, in which the teams have matches almost weekly. This could cause a reduction of the time spent on prevention strategies aimed to improve jump, speed, change-of-direction speed, and/or leg muscle strength [24], against increasing the tactical training time of match preparation. These specific training methods have been considered necessary to reduce physical deterioration [24] and the injury risk, especially of the ACL, and should not be eliminated [25]. Additionally, especially in such periods, deconditioning tests (e.g., a 20-m multistage fitness test, countermovement standing vertical jump, or 15-m sprint test), should be performed to identify physical decline and injury risk factors [26,27,28].

The high IR data of the ACL injury in our study is consistent with other studies carried out on the National Collegiate Athletic Association players (0.31/1000 h) [29]. Furthermore, it has previously been shown that ACL injury is more frequent and needs more recovery time in female soccer players than males [30] due to various anatomical, hormonal, neuromuscular, and biomechanical factors [31]. Although there are no data on the incidence in the Spanish male league of this injury, the differences between sexes are consistent if we compare it with the data shown in a male club of such a league (0.05/1000 h; 95%CI = 0.01, 0.21) [12]. Considering this significant problem, preventive strategies for ACL injury should be implemented in the exercise routine of female players, including strength exercises and muscle activation with biomechanics awareness of the lower limb [25].

In addition, the differences found between the circumstances with which the ACL and MCL injuries occur, where the first happens entirely in non-contact circumstances and the second, mostly with contact, provide new information about the incidents that cause such injuries. Although both injuries commonly have the same mechanism of injury (valgus, tibial external rotation, or a combination of both) [31], our findings indicate a difference in the event that causes this mechanism. In this way, prevention strategies should combine situations with and without contact to prevent injury to both structures. Even so, future research should examine specific injury mechanisms to optimally manage prevention programs.

Finally, the greater IRs in the non-contact circumstance of the hamstring and quadriceps strains with respect to contact is consistent with previous studies [32], showing similar IRs in total to other studies (0.9/1000 h) [2]. At the same time, the period when those injuries occurred with the highest IRs in the pre-season, is consistent with previous studies [32]. Preventive strategies have been shown to be useful, including increasing eccentric strength [33], flexibility gain of muscles of the thigh, a good balance of the concentric and eccentric strength of hip flexors and extensors, and correct core stability [34]. Regarding the periodization, Turner and Stewart [35] indicate the importance of strength training, especially in the off-and preseason, with exercises such as Romanian deadlifts, split squats, lateral lunges, or assisted Nordic curls, among others.

The greatest strength of this study was the study design, which followed the recommendations of the FIFA Consensus Statement [15]. In this statement, the strengths of injury cohort studies such as this one are collected in detail. This study also has several limitations. The first and main limitation was that this study was conducted only for one season with an average of eight players per team. Although the minimum participation per team was five players, ascertainment bias might exist because the injury reports can be influenced by the training and conditioning policies of each team. In addition, another limitation was that the data collection by survey was carried out at the end of the season; thus, our study has a recall bias. For this reason, we had to reject asking about the days lost or injury mechanisms since there was a possibility that this data was erroneous. For both reasons, we consider that the study should be repeated prospectively to register injury information over several seasons to collect important information (e.g., severity or mechanism of injury) and facilitate rate comparisons by increasing the number of cases.

The third limitation was the estimation of the exposure time. We used the time of exposure for training and matches with the monitoring of three individual teams, assuming such times for all teams were equal and may be overestimated. Although this method has been used by previous authors [16], we believe that future studies should use daily exposure times per player and record injuries and their characteristics weekly. Finally, another limitation of the study is that the data are not recent as they show the epidemiology of injuries for the 2016/2017 season.

Our study provides several clinical and research implications. In the first place, thanks to our study, the injury incidences of the first division Spanish women’s soccer players are examined for the first time, being essential to determine injury prevention strategies. Thus, preventive strategies should focus on ankle, knee, and thigh injuries. Finally, our findings have clarified in which period of the season the most frequent injuries of the lower limb occur and, in this way, enable knowing in which periods (i.e., the last two periods for the ankle/knee and preseason for the thigh) greater emphasis should be placed on the prevention strategies prescribed.

## 5. Conclusions

In conclusion, the greater injury IRs in the match with respect to the training in the first division Spanish women’s soccer players is similar to other first leagues of female soccer. Structures such as the ACL, meniscus, or ankle joint are most commonly injured in the non-contact circumstance. In addition, match situations involve a greater risk of injury than training, increasing the risk to the ankle and knee injuries as the season progresses. Therefore, prevention interventions for female players must take into account the factors that explain our findings (biomechanics of functional demands, specific muscle strength or cumulative fatigue, among others) and perform tasks directed at them. Future preventive strategies should focus on prospectively collecting specific injury mechanisms for several seasons.

## Figures and Tables

**Table 1 ijerph-18-03009-t001:** Player characteristics and exposure time in total and by game position in the first division Spanish women’s soccer players.

	ALL (N = 123)	GK (N = 13)	DF (N = 40)	MDF (N = 45)	FW (N = 25)
Age (years)	23.27 ± 4.07	22.92 ± 3.99	22.62 ± 3.90	23.24 ± 3.91	24.52 ± 4.62
First division debut (years)	18.01 ± 2.77	18.15 ± 3.44	17.72 ± 2.56	18.11 ± 2.69	18.20 ± 3.01
Body mass (kg)	59.5 ± 5.7	64.2 ± 6.9	60.2 ± 6.2	57.6 ± 4.5	59.5 ± 4.7
Stature (cm)	165.9 ± 5.7	168.8 ± 5.7	166.4 ± 1.1	164.6 ± 4.5	166.0 ± 5.5
BMI (kg/m^2^)	21.62 ± 1.56	22.56 ±2.5	21.72 ± 1.51	21.27 ± 1.19	21.61 ± 1.51
Dominant leg (r)	106	11	31	41	23
Exposure					
Total h/player	251.57 ± 43.75	266.91 ± 30.52	260.67 ± 43.42	247.03 ± 59.14	239.95 ± 24.26
Training h/player	222.50 ± 36.73	239.50 ± 19.29	228.87 ± 35.69	217.22 ± 48.38	214.37 ± 26.69
Pre-season	24.55 ± 10.35	32.31 ± 2.77	28.32 ± 6.44	24.06 ± 8.91	17.52 ± 13.77
Start of season	68.48 ± 18.46	73.94 ± 5.54	70.14 ± 17.23	63.88 ± 24.88	69.82 ± 14.97
Mid-season	56.95 ± 16.80	54.06 ± 11.71	61.27 ± 21.74	56.26 ± 16.89	53.77 ± 12.16
End of season	72.50 ± 12.73	78.19 ± 10.48	69.14 ± 12.91	73.01 ± 11.80	73.27 ± 14.81
Match h/player	29.07 ± 13.69	27.41 ± 17.16	31.79 ± 14.94	29.81 ± 16.48	25.58 ± 6.51
Pre-season	4.48 ± 1.82	5.51 ± 0.68	4.83 ± 1.44	4.20 ± 2.16	3.99 ± 2.05
Start of season	7.52 ± 5.73	7.46 ± 5.39	8.07 ± 6.29	8.17 ± 7.40	6.10 ± 2.58
Mid-season	8.09 ± 4.62	7.21 ± 6.42	9.61 ± 4.49	7.58 ± 5.40	7.23 ± 3.06
End of season	8.97 ± 5.46	7.22 ± 7.58	9.27 ± 6.22	9.85 ± 5.25	8.25 ± 4.39

GK: Goalkeeper; DF: Defender; MDF: Midfielder; FW: Forward; r: right; h: hours; BMI: Body mass index.

**Table 2 ijerph-18-03009-t002:** Injury frequencies and incidence rates according to training/match, type, and circumstance in the first division Spanish women’s soccer players in total and by position.

Injuries	All			Goalkeeper			Defender			Midfielder			Forward		
	N	(%)	IR (95% CI)	N	(%)	IR (95% CI)	N	(%)	IR (95% CI)	N	(%)	IR (95% CI)	N	(%)	IR (95% CI)
Total	113		3.65 (3.02, 4.37)	13	11.5	3.75 (2.08, 6.25)	32	28.3	3.07 (2.13, 4.28)	45	39.8	4.05 (2.99, 5.37)	23	20.4	3.83 (2.49, 5.66)
Training	50	44.2	1.70 (1.27, 2.22)	8	61.5	2.57 (1.19, 4.88)	15	46.9	1.64 (0.95, 2.64)	16	35.6	1.64 (0.97, 2.60)	11	47.8	2.05 (1.08, 3.57)
Match	63	55.8	19.02 (14.89, 23.97) *	5	38.5	14.04 (5.15, 31.13) *	17	53.1	13.38 (8.05, 20.98) *	29	64.4	21.63 (14.76, 30.65) *	12	52.2	18.78 (10.18, 31.93) *
Type															
Joint/ligament	64	56.6	2.07 (1.61, 2.62) †	9	69.2	2.59 (1.26, 4.76) †	19	59.4	1.82 (1.13, 2.79) †	25	55.6	2.25 (1.49, 3.27) †	11	47.8	1.83 (0.96, 3.19) †
Muscle/tendon	38	33.6	1.23 (0.88, 1.67) †	3	23.1	0.86 (0.02, 2.35)	9	28.1	0.86 (0.42, 1.58)	15	33.3	1.35 (0.78, 2.18)	11	47.8	1.83 (0.96, 3.19) †
Fracture/bone stress	11	8.8	0.35 (0.19, 0.62)	1	7.7	0.03 (0.01, 0.14)	4	12.5	0.38 (0.12, 0.92)	5	11.1	0.45 (0.16, 0.99)	1	4.3	0.17 (0.01, 0.08)
Circumstance															
Contact	45	39.8	1.45 (1.07, 1.93)	5	38.5	1.44 (0.53, 3.19)	12	37.5	1.15 (0.62, 1.96)	22	48.9	1.98 (1.27, 2.95)	6	26.1	1 (0.41, 2.08)
Non-contact	68	60.2	2.20 (1.72, 2.77)	8	61.5	2.31 (1.07, 4.38)	20	62.5	1.92 (1.20, 2.91)	23	51.1	2.07 (1.34, 3.06)	17	73.9	2.83 (1.71, 4.45)

N: frequencies of injuries; IR: incidence rate (per 1000 h of exposure); CI: confidence intervals. * Significant differences between training and match; *p* < 0.05. † Significant differences compared to fracture/bone stress; *p* < 0.01.

**Table 3 ijerph-18-03009-t003:** Injury frequencies and incidence rates according to localization and diagnosis in the first division Spanish women’s soccer players in total and by circumstance.

Localization/Diagnosis	All			Non-Contact			Contact		
	N	(%)	IR (95% CI)	N	(%)	IR (95% CI)	N	(%)	IR (95% CI)
Hip/groin	**5**	5.1	0.16 (0.06, 0.36) *	5	7.4	0.16 (0.06, 0.36)	-	-	-
Adductor strain	2	2	0.06 (0.01, 0.21)	2	2.9	0.06 (0.01, 0.21)	-	-	-
Abductor strain	1	1	0.03 (0.01, 0.15)	1	1.5	0.03 (0.01, 0.15)	-	-	-
Pubalgia	1	1	0.03 (0.01, 0.15)	1	1.5	0.03 (0.01, 0.15)	-	-	-
Bursitis	1	1	0.03 (0.01, 0.15)	1	1.5	0.03 (0.01, 0.15)	-	-	-
Thigh	24	24.5	0.78 (0.51, 1.13)	18	26.5	0.58 (0.35, 0.90)	6	13.3	0.19 (0.08, 0.40)
Hamstring strain	15	15.3	0.48 (0.28, 0.78)	11	16.2	0.35 (0.18, 0.62)	4	8.9	0.13 (0.04, 0.31)
Quadriceps strain	9	9.2	0.29 (0.14, 0.53)	7	10.3	0.23 (0.10, 0.45)	2	4.4	0.06 (0.01, 0.21)
Knee	29	29.6	0.94 (0.64, 1.32)	21	30.9	0.68 (0.43, 1.02)	8	17.8	0.26 (0.12, 0.49)
ACL’s Total rupture	11	11.2	0.35 (0.18, 0.62)	11	16.2	0.35 (0.18, 0.62)	-	-	-
MCL	11	11.2	0.35 (0.18, 0.62)	3	4.4	0.10 (0.02, 0.26)	8	17.8	0.26 (0.12, 0.49)
Distension	2	2	0.06 (0.01, 0.21)	-	-	-	2	4.4	0.06 (0.01, 0.21)
Partial rupture	7	7.1	0.23 (0.10, 0.45)	2	2.9	0.06 (0.01, 0.21)	5	11.1	0.16 (0.06, 0.36)
Total rupture	2	2	0.06 (0.01, 0.21)	1	1.5	0.03 (0.01, 0.15)	1	2.2	0.03 (0.01, 0.15)
Meniscus	7	7.1	0.23 (0.10, 0.45)	7	10.3	0.23 (0.10, 0.45)	-	-	-
Partial rupture	1	1	0.03 (0.01, 0.15)	1	1.5	0.03 (0.01, 0.15)	-	-	-
Total rupture	6	6.1	0.19 (0.08, 0.40)	6	8.8	0.19 (0.08, 0.40)	-	-	-
Lower leg	4	4.1	0.13 (0.04, 0.31) *	2	2.9	0.06 (0.01, 0.21)	2	4.4	0.06 (0.01, 0.21)
Calf strain	2	2	0.06 (0.01, 0.21)	2	2.9	0.06 (0.01, 0.21)	-	-	-
Laceration, Contusion	2	2	0.06 (0.01, 0.21)	-	-	-	2	4.4	0.06 (0.01, 0.21)
Ankle	30	30.6	0.97 (0.66, 1.37)	11	16.2	0.35 (0.18, 0.62)	19	42.2	0.61 (0.38, 0.94)
LCL	30	30.6	0.97 (0.66, 1.37)	11	16.2	0.35 (0.18, 0.62)	19	42.2	0.61 (0.38, 0.94)
Distension	16	16.3	0.52 (0.31, 0.82)	8	11.8	0.26 (0.12, 0.49)	8	17.8	0.26 (0.12, 0.49)
Partial rupture	10	10.2	0.32 (0.16, 0.58)	3	4.4	0.10 (0.02, 0.26)	7	15.6	0.23 (0.10, 0.45)
Total rupture	4	4.1	0.13 (0.04, 0.31)	-	-	-	4	8.9	0.13 (0.04, 0.31)
Foot/toe	6	6.1	0.19 (0.08, 0.40)	4	5.9	0.13 (0.04, 0.31)	2	4.4	0.06 (0.01, 0.21)
Fracture	2	2	0.06 (0.01, 0.21)	-	-	-	-	-	-
Stress fracture	2	2	0.06 (0.01, 0.21)	2	2.9	0.06 (0.01, 0.21)	-	-	-
Others	2	2	0.06 (0.01, 0.21)	2	2.9	0.06 (0.01, 0.21)	2	4.4	0.06 (0.01, 0.21)
Upper limb	10	66.7	0.32 (0.16, 0.58)	3	4.4	0.10 (0.02, 0.26)	7	15.6	0.23 (0.10, 0.45)
Hand/Elbow	4	26.7	0.13 (0.04, 0.31)	-	-	-	4	8.9	0.13 (0.04, 0.31)
Fracture	2	13.3	0.06 (0.01, 0.21)	-	-	-	2	4.4	0.06 (0.01, 0.21)
Others	2	13.3	0.06 (0.01, 0.21)	-	-	-	2	4.4	0.06 (0.01, 0.21)
Shoulder/Collar bone	6	40	0.19 (0.08, 0.40)	3	4.4	0.10 (0.02, 0.26)	3	6.7	0.10 (0.02, 0.26)
Fractures	2	13.3	0.06 (0.01, 0.21)	1	1.5	0.03 (0.01, 0.15)	1	2.2	0.03 (0.01, 0.15)
Others	4	26.7	0.13 (0.04, 0.31)	2	2.9	0.06 (0.01, 0.21)	2	4.4	0.06 (0.01, 0.21)
Trunk/Pelvis	5	5.1	0.16 (0.06, 0.36) *	4	4.4	0.13 (0.04, 0.31)	1	2.2	0.03 (0.01, 0.15)
Abdomen’s muscle strain	1	6.7	0.03 (0.01, 0.15)	1	1.5	0.03 (0.01, 0.15)	-	-	-
Low back pain	3	20	0.10 (0.02, 0.26)	3	4.4	0.10 (0.02, 0.26)	-	-	-
Pelvis fracture	1	6.7	0.03 (0.01, 0.15)	-	-	-	1	2.2	0.03 (0.01, 0.15)
TOTAL	113	100.0	3.65 (3.02, 4.37)	68	100.0	2.20 (1.72, 2.77)	45	100.0	1.45 (1.07, 1.93)

N: frequencies of injuries; IR: incidence rate (per 1000 h of exposure); CI: confidence intervals; ACL: anterior cruciate ligament; MCL: medial collateral ligament; LCL: lateral collateral ligament. * Significant differences compared to ankle, knee, and thigh; *p* < 0.01.

**Table 4 ijerph-18-03009-t004:** Injury frequencies and incidence rates of lower limb according to season’s period and training or match in the first division Spanish women’s soccer players.

Location	Pre-Season				Start of Season				Mid-Season				End of Season			
	Training		Match		Training		Match		Training		Match		Training		Match	
	N	IR (95% CI)	N	IR (95% CI)	N	IR (95% CI)	N	IR (95% CI)	N	IR (95% CI)	N	IR (95% CI)	N	IR (95% CI)	N	IR (95% CI)
Lower limb	5	1.66 (0.61, 3.67)	9	16.33 (7.97, 29.97) *	16	1.90 (1.12, 3.10)	7	7.58 (3.31, 14.99) *	14	2.00 (1.14, 3.27)	23	23.12 (15.01, 34.14) *	4	0.45 (0.14, 1.08)	17	15.41 (9.28, 24.18) *
Hip/groin	-	-	-	-	1	0.12 (0.01, 0.58)	1	1.08 (0.05, 5.34)	2	0.29 (0.05, 0.94)	1	1.01 (0.05, 4.96)	-	-	-	-
Thigh	3	0.99 (0.25, 2.70)	5	9.07 (3.32, 20.11) *	2	0.24 (0.04, 0.78)	1	1.08 (0.05, 5.34)	4	0.57 (0.18, 1.38)	4	4.02 (1.28, 9.70)	1	0.11 (0.01, 0.55)	3	2.72 (0.69, 7.40) *
Hamstring	2	0.66 (0.11, 2.19)	3	5.44 (1.38, 14.82)	1	0.12 (0.01, 0.58)	1	1.08 (0.05, 5.34)	2	0.29 (0.05, 0.94)	2	2.01 (0.34, 6.64)	1	0.11 (0.01, 0.55)	2	1.81 (0.30, 5.99)
Quadriceps	1	0.33 (0.02, 1.63)	2	3.63 (0.61, 11.99)	1	0.12 (0.01, 0.58)	-	-	2	0.29 (0.05, 0.94)	2	2.01 (0.34, 6.64)	-	-	1	0.91 (0.05, 4.47)
Knee	1	0.33 (0.02, 1.63)	2	3.63 (0.61, 11.99)	4	0.47 (0.15, 1.14)	1	1.08 (0.05, 5.34)	2	0.29 (0.05, 0.94)	10	10.05 (5.11, 17.91) *	1	0.11 (0.01, 0.55)	8	7.25 (3.67, 13.77) *
ACL	1	0.33 (0.02, 1.63)	1	1.81 (0.09, 8.95)	-	-	-	-	2	0.29 (0.05, 0.94)	2	2.01 (0.34, 6.64)	-	-	5	4.53 (1.66, 10.05)
MCL	-	-	1	1.81 (0.09, 8.95)	2	0.24 (0.04, 0.78)	1	1.08 (0.05, 5.34)	-	-	3	3.02 (0.77, 8.21)	1	0.11 (0.01, 0.55)	3	2.72 (0.69, 7.40)
Meniscus	-	-	-	-	2	0.24 (0.04, 0.78)	-	-	-	-	5	5.03 (1.84, 11.14)	-	-	-	-
Lower leg	1	0.33 (0.02, 1.63)	-	-	-	-	-	-	-	-	2	2.01 (0.34, 6.64)	-	-	1	0.91 (0.05, 4.47)
Ankle	-	-	1	1.81 (0.09, 8.95)	9	1.07 (0.52, 1.96)	4	4.33 (1.38, 10.44)	5	0.71 (0.26, 1.58)	5	5.03 (1.84, 11.14) *	1	0.11 (0.01, 0.55)	5	4.53 (1.66, 10.05) *
Foot	-	-	1	1.81 (0.09, 8.95)	-	-	-	-	2	0.29 (0.05, 0.94)	2	2.01 (0.34, 6.64)	1	0.11 (0.01, 0.55)	-	-

N: frequencies of injuries; IR: incidence rate (per 1000 h of exposure); CI: confidence intervals; ACL: anterior cruciate ligament; MCL: medial collateral ligament. * Significant differences between training and match (*p* = 0.05).

## Data Availability

The data presented in this study are available on request from the corresponding author. The data are not publicly available due to privacy reasons.

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
