# Peer review of "Epidemiology of Injuries in First Division Spanish Women’s Soccer Players"

_ijerph, 2021, doi:10.3390/ijerph18063009_

Round 1
Reviewer 1 Report
The study Epidemiology of injuries in first division Spanish women's soccer players is an interesting paper and could be very informative to the readers of International Journal of Environmental Research and Public Health. Thus, I would encourage the authors to polish the study.
The introduction gives the reader a good overview of the currently published data. The authors manages to work out that data on epidemiology of injuries in female soccer are limited in literature. Thus the introduction leads stringently to the research problem and the first part of research question (examine the injury frequencies 51 and IRs of the first division Spanish women's soccer players). However, second part of the research question should be taken up more precisely in the introduction (and referenced), analogous to the variable training/match.
The methodology chapter is suitable to answer the research questions posed. The reader can understand the methodological approach of the author. Please explain why the 2016/2017 season is analysed and not a more recent one. If necessary, please include this in the limits of the studies. However, the chapter on analysis needs an extension of the description of the statistical procedure. Please add description how you calculated significant differences. Please think also about adding effects sizes. Please consider the study-wise error and think about the adjustment by the false discovery rate.
The presentation of the results is appropriate. Please indicate the statistical parameters of calculations (F=, t=, p=, ….). Please define what you mean by significant. Please also add this to the tables.
The discussion is structured and discusses own data appropriately, taking into account the literature.
Author Response
Response to Reviewer 1 Comments
We thank the reviewer for their feedback. We provide a point-by-point answer to each specific comment. In addition, changes at the manuscript are provided with the track-changes mode.
Comments:
The study Epidemiology of injuries in first division Spanish women's soccer players is an interesting paper and could be very informative to the readers of International Journal of Environmental Research and Public Health. Thus, I would encourage the authors to polish the study.
Authors’ response: We want to thank the reviewer 1 the consideration for the manuscript. Thank you very much for your comments.
The introduction gives the reader a good overview of the currently published data. The authors manages to work out that data on epidemiology of injuries in female soccer are limited in literature. Thus the introduction leads stringently to the research problem and the first part of research question (examine the injury frequencies 51 and IRs of the first division Spanish women's soccer players). However, second part of the research question should be taken up more precisely in the introduction (and referenced), analogous to the variable training/match.
Authors’ response: Thank you very much for your suggestion. We agree with you. We have created a gap in relation to the importance of the analysis of the factors examined in this study to subclassify the incidence rates by game position, period or circumstance. We consider that this way, the second part of the objective is reinforced.
The methodology chapter is suitable to answer the research questions posed. The reader can understand the methodological approach of the author. (a) Please explain why the 2016/2017 season is analysed and not a more recent one. If necessary, please include this in the limits of the studies. (b) However, the chapter on analysis needs an extension of the description of the statistical procedure. Please add description how you calculated significant differences. Please think also about adding effects sizes. Please consider the study-wise error and think about the adjustment by the false discovery rate.
Authors’ response: (a) Thank you very much for your suggestion. Yes, we agree, it could have been a more current analysis. The time in collecting data, its analysis, and writing of the manuscript, added to the COVID year, has supposed a delay in its proposal for publication. We think that this situation is more common than desired. For example, the study by Noya et al. on Epidemiology of injuries in First Division Spanish football in men, was published in 2014 and its data are in relation to the 2008/2009 season. We have added this limitation to the manuscript. (b) Thank you very much for your suggestions. We have expanded the information on how significant differences were calculated. On the other hand, we have added effect sizes calculations as you suggest, using the risk ratio as the effect size statistic as suggested by previous authors. Finally, we agree, in those cases where multiple comparisons are made, as in the type of injury in Table 2, adjustments should be made. We have used the 99% IC comparison as an adjustment for multiple comparisons. This has been explained in the statistical analysis section and made the corresponding modifications in the results section.
The presentation of the results is appropriate. Please indicate the statistical parameters of calculations (F=, t=, p=, ….). Please define what you mean by significant. Please also add this to the tables.
Authors’ response: Thank you so much for your comments. Since we use the 95% CI overlap, the parameters F and t cannot be obtained. We have defined the significance value of p in the tables.
The discussion is structured and discusses own data appropriately, taking into account the literature.
Authors’ response: Thank you for your feedback to improve our work.

Reviewer 2 Report
- I suggest the mean age of the players participating in the study be included in the method section
- It could interest some readers to know how many players each team was eligible to register for the season. How many who participated in the study were regular players in the respective team
- The reliability and validity of the questionnaire used to collect the data could interest some readers
- It is not clear if the anthropometric measurements were recalled or measured and what their reliability and validity was in line 72-72.
- Concussion is one major injury in male soccer match the current report is silent about it in women soccer line 72-73.
- The mean BMI of defenders is in the overweight category. One wonder if the author have investigated the diet and the training program of these professional players
- In the method section I suggest the author include a section describing how often league matches were scheduled for each team and the general picture of the training period/interval before the game for each club.
- One wonders as to how long these players who are part of the study have been turned professional before they were enrolled in the study.
Author Response
Response to Reviewer 2 Comments
We thank the reviewer for their feedback. We provide a point-by-point answer to each specific comment. In addition, changes at the manuscript are provided with the track-changes mode.
Comments:
- I suggest the mean age of the players participating in the study be included in the method section
Authors’ response: Thank you for your comment. We have added this information.
- It could interest some readers to know how many players each team was eligible to register for the season. How many who participated in the study were regular players in the respective team
Authors’ response: Thank you very much for your comment. We agree with you, this would be interesting. Given that the number of players on each team is specific, there are even drop-offs in the middle of the season, giving this information is not possible. We reference this information by indicating that our sample represents 44% of the league's players, to reflect the information that you comment in an estimative way.
- The reliability and validity of the questionnaire used to collect the data could interest some readers
Authors’ response: Thank you very much for your comment. As indicated in the methods, our questionnaire follows that proposed by Fulled et al. in his "Consensus statement on injury definitions and data collection procedures in studies of football (soccer) injuries". This questionnaire, developed specifically for FIFA, and its use in authors who are going to study epidemiology in soccer, lacks a validity and reliability study, possibly due to its complexity and multiple items of different aspects. What we have done, responding to your suggestion and to show that our study complies with what is proposed in this consensus, is to fill in table 5 of the consensus, which is a checklist for this type of study. We attach as supplementary material the table so that reviewers can confirm this.
- It is not clear if the anthropometric measurements were recalled or measured and what their reliability and validity was in line 72-72.
Authors’ response: Thank you very much for your comment. We have specified that this information was in the retrospective questionnaire.
- Concussion is one major injury in male soccer match the current report is silent about it in women soccer line 72-73.
Authors’ response: Thank you very much for your comment. As indicated in ln 105-109, "In this way and as the statement suggests, we grouped locations and types with small frequencies, using finally 8 locations (Hip / groin, thigh, knee, lower leg, ankle, foot / toe, upper limb and, trunk / pelvis) and 3 types (Joint / ligament, muscle / tendon, and fracture / bone stress) of injury ". We obtained expected results in relation to concussion, as something with a very low incidence, as found by Noya et al. for his article "Epidemiology of injuries in First Division Spanish football" in men (concussion is the injury with the lowest incidence; 0.04). Thus, in our case, it is not specified because it is grouped. In addition, as lacerations, which do not require specific rehabilitation, concussions are usually pathologies affected by recall bias.
- The mean BMI of defenders is in the overweight category. One wonder if the author have investigated the diet and the training program of these professional players
Authors’ response: Thank you very much for your comment. We have identified an error. Two players from that position (defense), in the database, were 62 and 65 cm tall, instead of 162 and 165 cm, respectively. This influenced both the height and the BMI, hence for the DF, there was an excessive SD. We have corrected this. Now, the DF data are similar to the rest of the positions in both variables. We have also corrected this for the total of the players both in the table and in the text.
- In the method section I suggest the author include a section describing how often league matches were scheduled for each team and the general picture of the training period/interval before the game for each club.
Authors’ response: Thank you very much for your suggestion. We have added the matches corresponding to the league and their sub-classification by period. As for the weekly training interval, this is specific to each club and each week, being determined by the weekly planning.
- One wonders as to how long these players who are part of the study have been turned professional before they were enrolled in the study.
Authors’ response: Thank you very much for your question. Yes, we agree with you, this information is valuable. We decided to indicate the years with which they debuted in the first division. Indirectly, by subtracting from their age, the age at which they debuted, it is possible to estimate how many years they can have been in the first division.

Round 2
Reviewer 1 Report
Many thanks to the authors for the revision. The revision has increased the quality of the manuscript. The manuscript is ready for publication. Congratulations to the authors.
Reviewer 2 Report
none